# Anti-epileptic drug topiramate upregulates TGFβ1 and SOX9 expression in primary embryonic palatal mesenchyme cells: Implications for teratogenicity

**Syed K. Rafi**[1]*, **Jeremy P. Goering**[1], **Adam J. Olm-Shipman**[1], **Lauren A. Hipp**[1], **Nicholas J. Ernst**[1], **Nathan R. Wilson**[1¤a], **Everett G. Hall**[1¤b], **Sumedha Gunewardena**[2], **Irfan Saadi**[1]*

**1** Department of Anatomy and Cell Biology, University of Kansas Medical Center, Kansas City, Kansas, United States of America, **2** Department of Molecular and Integrative Physiology, University of Kansas Medical Center, Kansas City, Kansas, United States of America

¤a Current address: Center for Regenerative Medicine, Massachusetts General Hospital, Harvard Medical School, Boston, Massachusetts, United States of America
¤b Current address: Clinical Research Training Center, Institute of Clinical and Translational Sciences, Washington University, St. Louis, Missouri, United States of America
* rafigene@yahoo.com (SKR); isaadi@kumc.edu (IS)

**Data Availability Statement:** All relevant data are within the manuscript and its Supporting Information files.

## Abstract

Topiramate is an anti-epileptic drug that is commonly prescribed not just to prevent seizures but also migraine headaches, with over 8 million prescriptions dispensed annually. Topiramate use during pregnancy has been linked to significantly increased risk of babies born with orofacial clefts (OFCs). However, the exact molecular mechanism of topiramate teratogenicity is unknown. In this study, we first used an unbiased antibody array analysis to test the effect of topiramate on human embryonic palatal mesenchyme (HEPM) cells. This analysis identified 40 differentially expressed proteins, showing strong connectivity to known genes associated with orofacial clefts. However, among known OFC genes, only TGFβ1 was significantly upregulated in the antibody array analysis. Next, we validated that topiramate could increase expression of TGFβ1 and of downstream target phospho-SMAD2 in primary mouse embryonic palatal mesenchyme (MEPM) cells. Furthermore, we showed that topiramate treatment of primary MEPM cells increased expression of SOX9. SOX9 overexpression in chondrocytes is known to cause cleft palate in mouse. We propose that topiramate mediates upregulation of TGFβ1 signaling through activation of γ-aminobutyric acid (GABA) receptors in the palate. TGFβ1 and SOX9 play critical roles in orofacial morphogenesis, and their abnormal overexpression provides a plausible etiologic molecular mechanism for the teratogenic effects of topiramate.

## Introduction

Topiramate is an anti-epileptic drug that was approved by the U.S. Food and Drug Administration (FDA) for the treatment of partial onset or primary generalized tonic-clonic seizures in

**Funding:** This project was supported in part by the National Institutes of Health Center of Biomedical Research Excellence (COBRE) grant (National Institute of General Medical Sciences P20 GM104936 and P30 GM122731, I.S.), Kansas IDeA Network for Biomedical Research Excellence grant (National Institute of General Medical Sciences P20 GM103418, I.S.), Kansas Intellectual and Developmental Disabilities Research Center grant (U54 Eunice Kennedy Shriver National Institute of Child Health and Human Development, HD090216, I.S. and S.G.), and a National Institute of Dental and Craniofacial Research grant (DE026172, I.S.).

**Competing interests:** The authors do not have any competing financial interests pertaining to the studies presented here.

1996 and for migraine prophylaxis in 2004. Topiramate is currently used either as a monotherapy or an adjunctive therapy to treat migraines, partial-onset seizures, primary generalized tonic-clonic seizures, and other seizure disorders. In addition, topiramate has been used off-label for binge-eating disorder [1], bulimia nervosa [2], alcohol use disorder [3], anti-psychotic induced weight gain [4], and essential tremor [5]. Verispan's Vector One®: National (VONA) and IQVIA's Total Patient Tracker® (TPT) reported that, between January 2007 and December 2010, approximately 4.3 million patients filled 32.3 million topiramate prescriptions (FDA Drug Safety Communication, 03-04-2011). This number has risen to approximately 4.1 million patients receiving a prescription for topiramate between March 2014 and February 2016 (FDA Pediatric Post-marketing Pharmacovigilance and Drug Utilization Review for topiramate; June 20, 2016). According to estimates in the United States alone, approximately 1.3 million epileptic women are of childbearing age [6], and approximately 24,000 children are born annually to epileptic mothers (North American Antiepileptic Drug Pregnancy Registry) [7]. Thus, control of seizures during pregnancy is an important public health challenge, as women with epilepsy have a higher risk of peripartum complications including stillbirth, preeclampsia, preterm labor, and a 10-fold increase in mortality compared to women without epilepsy [8]. Importantly, in a recent study [9], twice as many women with migraines (46%) were prescribed topiramate than women with epilepsy or seizures (20%), exposing an even larger proportion of women of childbearing age to potential topiramate teratogenicity.

Topiramate use during pregnancy has been linked to a significantly increased risk of birth defects affecting orofacial, cardiac and urogenital development [10–12]. Multiple studies have reported that the incidence of oral clefts in particular is increased in topiramate-exposed pregnancies [10, 11, 13, 14]. In the North American Antiepileptic Drug Pregnancy Registry [15], the relative risk of oral clefts in topiramate-exposed pregnancies was ~13-fold higher than the general risk in births [16]. The FDA has now changed the classification of topiramate from a pregnancy-C category to a pregnancy-D category drug, warning that topiramate can cause fetal harm when administered to a pregnant woman (FDA Drug Safety Communication, 03-04-2011). Despite broad use and teratogenic potential of topiramate, the molecular mechanism(s) underlying the increased occurrence of major congenital malformations is not understood [6, 7, 11, 13, 17].

We sought to identify molecular clues to the teratogenic effect of topiramate on palate development. Normal palate development during embryogenesis is a multistep process that begins with bilateral vertical growth of the palatal shelves adjacent to the tongue till embryonic day 13.5 (E13.5). These shelves then elevate above the tongue, move horizontally and fuse in the midline to form the palate by E15.5. The palatal shelf is mainly composed of the oral ectoderm derived epithelial cells and the neural crest derived mesenchymal cells. Defects in neural crest function are sufficient to cause cleft palate [18, 19]. To investigate the potential effect of topiramate exposure on palate development, we first treated human embryonic palate mesenchyme (HEPM) cells with a high dose of topiramate, then performed an unbiased exploratory antibody-array approach to identify misregulated proteins. Our analysis showed upregulation of transforming growth factor beta one (TGFβ1) expression and altered activation of cell survival networks. We then validated our findings in primary mouse embryonic palate mesenchyme (MEPM) cells following exposure to a range of topiramate concentrations. We showed that TGFB1 levels are indeed upregulated even at physiological 50μM topiramate treatment for six hours. Topiramate treatment of primary MEPM cells also increased expression of SRY-Box Transcription Factor 9 (SOX9), a TGFB1 target that causes cleft palate when abnormally expressed. We propose that perturbation of TGFβ pathway and SOX9 expression through γ-aminobutyric acid (GABA) receptors in the embryonic palate represents a plausible etiologic mechanism underlying topiramate-induced oral clefts.

## Materials and methods

### Culture of HEPM cell line

Human embryonic palatal mesenchyme (HEPM) cells, available commercially (ATCC, CRL-1486), were cultured in DMEM media, with high concentration of glucose and pyruvate, supplemented with 10% fetal bovine serum and penicillin/streptomycin. The HEPM cells were kept at low passage numbers and inspected visually for signs of differentiation. Topiramate (Sigma-Aldrich, St. Louis, MO) was resuspended in ethanol. HEPM cells were treated with 1000μM topiramate and cultured at 37˚C for 6 hours. Cells were briefly washed with PBS, scraped and flash-frozen for subsequent RNA and protein extraction.

### Isolation and culture of primary MEPM cells

Palatal shelves were excised from wildtype E13.5 mouse embryos. Briefly, the embryo heads were cut, followed by removal of the lower jaw and tongue, thus exposing the palatal shelves. The palatal shelves were subsequently dissected, and the cells dissociated into a single-cell suspension by incubating the palatal shelves in 0.25% Trypsin for 10 minutes and then vigorously pipetting up and down to mechanically separate the cells. The resulting Mouse Embryonic Palatal Mesenchyme (MEPM) cells were then cultured in Dulbecco's Modified Eagle Medium (DMEM) with high glucose and supplemented with 10% Fetal Bovine Serum and Penicillin/Streptomycin. MEPMs were used fresh, never-passaged, before drug treatment. MEPM cells were treated with 25, 50 or 100 μM topiramate (APExBio, Houston, TX) and incubated for 6 hours at 37˚C. GABA inhibitor (10 μM), flumazenil (APExBio, Houston, TX), was added with 50 μM topiramate for the 6 hour treatment duration. Ethanol was used for vehicle treatment. All experiments involving animals were carried out with a protocol approved by the University of Kansas Medical Center (KUMC) Institutional Animal Care and Use Committee, in accordance with their guidelines and regulations (Protocol Number: 2018–2447). The experiments reported in this study are from 12 timed-pregnant C57BL/6J female mice (The Jackson Laboratory, Bar Harbor, ME) that were euthanized using a $CO_2$ chamber followed by harvesting of E13.5 embryos.

### Antibody array analysis

Protein extracts from HEPM cells with vehicle (Control) or with topiramate treatment (1000 μM for 6 hours) were analyzed by Full Moon BioSystems (Sunnyvale, CA) Cell Signaling Explorer antibody arrays (SET100), according to manufacturer's protocol. Briefly, control and topiramate-treated HEPM cell lysates were labeled with Cy3 fluorophore using the antibody array assay kit from Full Moon BioSystems (KAS02). Antibody array slides were independently incubated with labeled lysates, washed, and scanned using microarray scanner from Agilent (Santa Clara, CA).

### Statistical analysis of antibody array data

Each antibody array comprised of two technical replicates. The experiment was performed in biological duplicates for downstream statistical analysis. Each probe from each of the biological and technical replicates was first determined to be significantly expressed relative to its background intensity. This significance calculation was based on a $\leq 0.05$ cutoff of the Benjamini and Hochberg [20] adjusted p-value of the t-statistic of the difference between the mean expression (F532 mean) and mean background (B532 mean) intensities. A gene was regarded as not significantly expressed under a particular treatment if any one of its biological replicates was not significantly expressed. The first filtering step removed all genes that were not

significantly expressed in either one of the treatment groups. The remaining probes were background corrected using a modified Robust Multi-array Average (RMA) [21] algorithm for protein arrays. The background adjusted probes were log transformed (base 2) and quantile normalized. Technical replicates were then averaged (geometric average) to give the background adjusted normalized expression for each biological replicate. A two-way ANOVA model (factor 1: treatment with levels Control and topiramate treated; factor 2: antibody array with levels array 1 and array 2) was fitted to the data to determine the gene level significance of the difference in expression between Control and topiramate treated samples. Protein expression with a p-value ≤ 0.05 and absolute fold change ≥ 1.15 were deemed sufficiently differentially expressed for protein arrays, yielding 40 proteins for further analysis.

## Gene interaction analysis

The Ingenuity Pathway Analysis software (IPA, Qiagen, www.qiagenbioinformatics.com) was used to build a gene interaction network of the 40 differentially expressed proteins and 107 known genes associated with orofacial clefts identified by IPA. The aim was to establish a single network of interactions between as many of the 40 differentially expressed proteins from the antibody array and the 107 genes associated with orofacial clefts. IPA performs this task with the aid of its knowledge database consisting of literature-based curated information on genes and gene product interactions.

## Quantitative PCR analysis

Quantitative PCR (qPCR) analysis was used to test expression of the *TGFB1* in HEPM cells treated with 1mM Topiramate for 6 hours and *Tgfb1* in MEPM cells treated with 50μM Topiramate for 6 hours. RNA was extracted using NucleoSpin RNA XS kit (Takara, Kusatsu, Japan). 1 μg of RNA from each sample was used to generate cDNA with qScript cDNA SuperMix (QuantaBio, Beverly, MA). *OAZ1 and B2m* housekeeping genes were used to normalize data for HEPM and MEPM cells, respectively. Analysis was performed on 4–5 sets of biological replicates, each with 2 technical replicates per gene. Statistical significance was calculated using a Student's t-test. Primer sequences are listed in S1 Fig.

## Western blotting

For protein extraction, MEPM cells were briefly washed with PBS, scraped and either flash-frozen or lysed immediately. Cells were lysed by suspension in radioimmunoprecipitation assay (RIPA) buffer with HALT protease inhibitor Cocktail (Thermo Scientific, Waltham, MA) and by agitation for 30 minutes at 4˚C. Cell lysates were centrifuged for 10 minutes at 13,000 rcf and the protein extracts (supernatant) collected. Lysates were then electrophoresed in 4–15% gradient Mini-Protean TGX Stain-Free precast gels (Bio-RAD, Hercules, CA). After electrophoresis, the gels were exposed to UV light for 45 seconds to develop the total protein signal and imaged on a ChemiDoc System (Bio-RAD, Hercules, CA) before being transferred onto Immobilon PVDF membranes (EMD Millipore, Billerica, MA). PVDF membranes were then blocked in Odyssey Blocking Buffer (Li-Cor, Lincoln, NE) either overnight at 4˚C or at room temperature for 1 hour. Primary antibodies used were anti-TGFβ1 (1:1000; Abcepta, AP12348A, Cambridge, MA), anti-phospho-SMAD2 (1:5000; Cell Signaling Technologies, 3108, Danvers, MA), anti-SMAD2 (1:5000; Cell Signaling Technologies, 5339, Danvers, MA) and anti-SOX9 (1:5000; Abcepta, AM1964b, San Diego, CA), and anti-SOX10 (1:5000; Aviva Systems, ARP33326, San Diego, CA). Secondary antibodies used were HRP-linked goat anti-rabbit IgG (1:10,000; Cell Signaling Technologies, Danvers, MA) and HRP-linked goat anti-mouse IgG (1:10,000; Santa Cruz Biotechnologies, Dallas, TX). Femto SuperSignal West ECL

reagent (Thermo Scientific, Waltham, MA) was used to develop the signal. Image Lab software (Bio Rad, Hercules, CA) was used to quantitate total protein and western blot intensity. Each blot was normalized to the total protein loaded, and then fold change calculated by dividing total drug-treated samples by vehicle-treated sample.

### Immunofluorescence and imaging analysis

MEPM cells, cultured as described above, were fixed in 4% paraformaldehyde (PFA) for 10 min, blocked in phosphate buffered saline (PBS) with 1% goat serum and 0.1% Tween, and stained using Anti-TGFβ1 (1:1000; Abcam, Cambridge, MA). After staining, coverslips were mounted in containing DAPI (Vector Labs, Burlingame, CA). Individual cells were imaged, and the levels of TGFB1 fluorescence were quantitated in at least 30 cells per treatment from 3 independent experiments using NIH ImageJ software. Briefly, we used NIH ImageJ to calculate the corrected total cell fluorescence (CTCF) in each cell, using the formula: CTCF = Integrated Density–(Area of selected cell x Mean fluorescence of background readings).

## Results

### Antibody-array-based analysis of HEPM cells following topiramate treatment

Protein extracts from HEPM cells with supra-physiological topiramate treatment (1000 μM for 6 hours) or without the treatment (Control) were assayed by Full Moon BioSystems (Sunnyvale, CA) Cell Signaling Explorer antibody-array. The Cell Signaling Explorer array includes antibodies for 1358 individual proteins, in two technical replicates, encompassing 20 cellular pathways. The antibody array experiment was performed with two biological replicates. The results were analyzed for statistical significance as described in the Materials and Methods section. Protein levels of 57 gene products were significantly altered (p<0.05, S1 Table). We used a 1.15-fold cutoff (|FC|≥1.15) [22, 23], resulting in 40 differentially expressed proteins with 19 proteins downregulated and 21 upregulated (Table 1). To determine the importance of these 40 altered gene products to orofacial morphogenesis and OFCs, we used Ingenuity Systems pathway analysis (IPA) as well as manual literature curation.

### IPA analysis of proteins with altered expression following topiramate treatment shows connectivity with genes associated with orofacial clefts

The antibody array analysis identified statistically significant changes in expression of 40 proteins following topiramate treatment (Table 1; p<0.05, |FC|≥1.15). Separately, IPA software reported 107 known genes associated with orofacial clefts (S2 Table). Only 18 of the 107 OFC genes were represented on the Full Moon BioSystems Cell Signaling Explorer antibody array. Among these 18 proteins encoded by OFC genes, only TGFβ1 was significantly altered (increased) upon topiramate treatment of HEPM cells.

We also analyzed the 40 gene products with IPA for changes in diseases and bio-functions (S3 Table). The top predicted categories were related to "cell death and survival", and "organismal growth and development". Overall, IPA predicted that topiramate treatment decreased cell viability (S3 Table; IPA Ranks 13, 29) and increased apoptosis (S3 Table; IPA Rank 37). However, IPA also correctly predicted increased cell viability of neurons (S3 Table; IPA Rank 7) [24], consistent with a cell-type specific effect of topiramate. The top IPA predicted networks also indicated an effect on cell growth and survival pathways (S4 Table). Next, we looked at connectivity of the genes encoding the 40 altered proteins to the IPA-reported 107 genes

**Table 1. List of 40 proteins with statistically significant change in expression and a 1.15-fold-change cut-off in HEPM cells following topiramate treatment.**

| Protein Name | Symbol | UniProtKB ID | Antibody Array ID | Fold Change |
|---|---|---|---|---|
| Checkpoint kinase 2 | CHEK2 | O96017 | 659 | 1.96 |
| Low-density lipoprotein receptor class A domain-containing protein 1 | LDLRAD1 / LRP1 | Q5T700 | 109 | 1.64 |
| Aldehyde dehydrogenase 7 | ALDH3B1 | P43353 | 92 | 1.46 |
| NADPH oxidase 5 | NOX5 | Q96PH1 | 282 | 1.46 |
| Proto-oncogene tyrosine-protein kinase receptor Ret | RET | P07949 | 490 | 1.42 |
| Cardiac troponin I | TNNI3(cTnI) | P19429 | 146 | 1.39 |
| Transforming Growth Factor beta 1 | TGFB1 | P01137 | 377 | 1.39 |
| Unconventional myosin Id | MYO1D | O94832 | 276 | 1.37 |
| Tyrosine-protein kinase receptor 3 | TYRO3 | Q06418 | 478 | 1.33 |
| Cyclosome 1 | APC1 | Q9H1A4 | 1093 | 1.32 |
| Superoxide dismutase 1 | SOD1 | P00441 | 996 | 1.31 |
| Oral cancer-overexpressed protein 1 | ORAV1 / ORAOV1 | Q8WV07 | 1078 | 1.30 |
| Growth arrest and DNA damage-inducible proteins-interacting protein 1 | GADD45GIP1 | Q8TAE8 | 1285 | 1.27 |
| Phosphatidylinositol-glycan biosynthesis class H protein | PIGH | Q14442 | 453 | 1.25 |
| CD30 Ligand | CD153 | P32971 | 220 | 1.24 |
| Patched | PTCH1 | Q13635 | 35 | 1.23 |
| Cadherin-10, T2-cadherin | CDH10 | Q9Y6N8 | 415 | 1.21 |
| ATP synthase subunit delta, mitochondrial | ATP5D | P30049 | 261 | 1.19 |
| TNF Receptor1-associated DEATH domain protein | TRADD /TNFR1 | Q15628 | 716 | 1.19 |
| Alcohol dehydrogenase class 4 mu/sigma chain | ADH7 | P40394 | 91 | 1.19 |
| Interferon Regulatory Factor 4 | IRF4 | Q15306 | 730 | 1.18 |
| Bcl10-interacting CARD protein | C9ORF89 | Q96LW7 | 422 | -1.15 |
| S-phase kinase-associated protein 1 | SKP1A/p19 | P63208 | 363 | -1.16 |
| B melanoma antigen 3/ Cancer/testis antigen 2.3 | BAGE3 | Q86Y29 | 420 | -1.20 |
| Cancer-associated Gene Protein | ATBP3 / MTUS1 | Q7Z7A3 | 907 | -1.20 |
| SUMO1-activating enzyme 2 | UBA2 | Q9UBT2 | 874 | -1.22 |
| Tissue inhibitor of metalloproteinases 2 | TIMP2 | P16035 | 537 | -1.22 |
| RING finger and WD repeat domain protein 2 | RFWD2 | Q8NHY2 | 1107 | -1.23 |
| Coagulation factor VII (light chain, Cleaved-Arg212) | FA7 | P08709 | 1156 | -1.29 |
| Rho guanine nucleotide exchange factor 3 | ARHGEF3 | Q9NR81 | 1123 | -1.30 |
| Cytochrome c oxidase assembly protein COX11 | COX11 | Q9Y6N1 | 911 | -1.31 |
| Diacylglycerol Kinase eta | DGKH | Q86XP1 | 908 | -1.36 |
| ATP-binding cassette sub-family A member 8 | ABCA8 | O94911 | 848 | -1.37 |
| Shugoshin-like 1 | SGOL1 | Q5FBB7 | 895 | -1.37 |
| Guanylate Cyclase Beta 1 | GUCY1B3 | Q02153 | 853 | -1.38 |
| Cell division cycle 7 related kinase | CDC7 | O00311 | 1044 | -1.38 |
| Aldo-keto reductase family 1 member C-like protein 1 | AKR1CL1 | Q5T2L2 | 246 | -1.41 |
| Interleukin 2 | IL2 | P60568 | 832 | -1.55 |
| Cytochrome P450 7B1 | CYP7B1 | O75881 | 802 | -1.82 |
| Ubiquitin-protein ligase E3B | UBE3B | Q7Z3V4 | 291 | -2.19 |

associated with orofacial clefts (Fig 1). TGFβ1 was the only OFC gene among the 40 topiramate-altered proteins in our analysis. A single network was able to connect 22 (9 downregulated, 13 upregulated) of these 40 (55%) genes encoding differentially expressed proteins to 87 OFC genes (81%), either directly or indirectly. TGFβ1 showed the highest number of connections to known genes associated with orofacial clefts (Fig 1). Therefore, we focused our validation studies on TGFβ1.

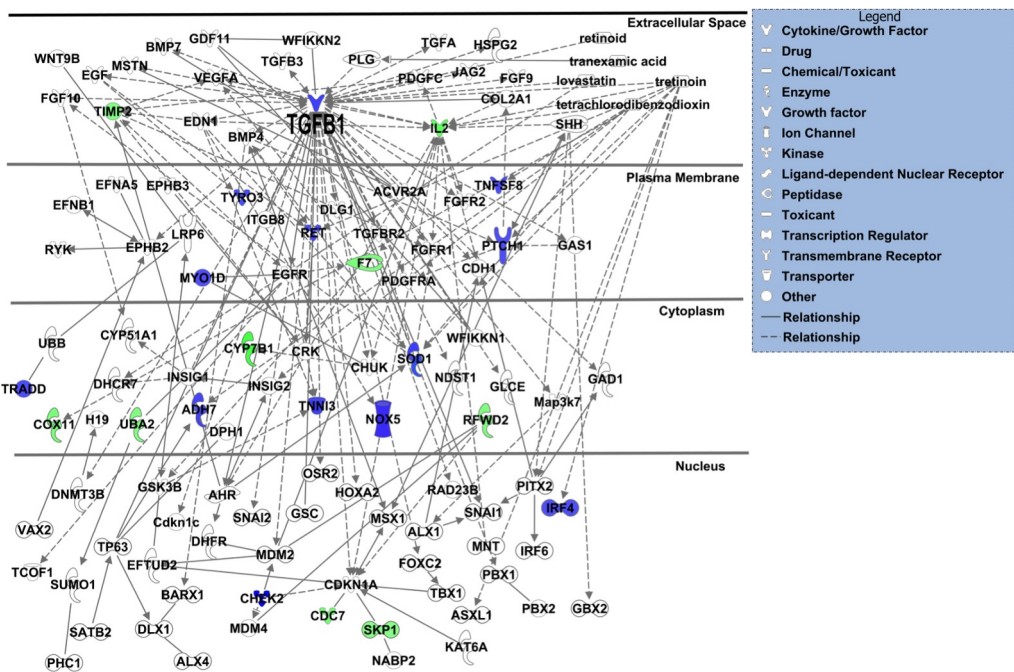

**Fig 1. TGFβ1 showed highest connectivity in IPA-generated network of differentially expressed gene products from topiramate-treated HEPM cells and known orofacial clefting-associated genes.** The 40 gene products with differential expression following 6-hour, 1000μM topiramate treatment of HEPM cells were analyzed with Ingenuity Systems pathway analysis (IPA) software for possible interaction with 107 IPA-identified genes associated with orofacial clefts. A single resulting network accounted for 22 (55%) of the topiramate-treated HEPM genes in association with 87 (81%) known OFC-related genes. The upregulated or downregulated genes from the HEPM data were colored blue or green, respectively. The only gene common between the two datasets was *TGFB1*, which also displays the highest connectivity in the network.

## Validation of TGFβ1 upregulation in primary Mouse Embryonic Palate Mesenchyme (MEPM) cells following treatment with physiological levels of topiramate

Our analysis indicated TGFβ1 as the central molecule affected by topiramate treatment of HEPM cells. To confirm that our finding was not an artifact of using an immortalized cell line and supraphysiological levels of topiramate, we used primary MEPM cells and physiological topiramate doses. The peak serum level for even topiramate monotherapy is approximately 25–50 μM [25–27]. Therefore, we initially included both the physiological 50 μM concentration and two supraphysiological concentrations to treat fresh primary MEPM cells from E13.5 embryonic palates for 6 hours (Fig 2A–2E). Following treatment, cells were fixed and immunostained using an anti-TGFB1 antibody (Fig 2A–2D). Individual cells were imaged, and the levels of TGFB1 were quantitated using NIH ImageJ software. The experiment was repeated three times with different primary MEPM isolations. We saw a significant increase in intracellular TGFβ1 level at the physiological 50μM topiramate treatment (Fig 2E).

We also performed Western blot analysis following 25μM and 50μM topiramate treatments (Fig 2F) and showed a significant increase with 50μM topiramate (Fig 2G). There was no increase in *TGFB1* (HEPM) or *Tgfb1* (MEPM) mRNA transcripts at the end of the 6-hour topiramate treatment (S1 Fig). To test for an upregulation of the TGFβ1 signaling pathway, we also assessed levels of phospho-SMAD2 (P-SMAD2), a downstream effector molecule (transcription factor) of the ligand-bound TGFβ receptor that requires phosphorylation to

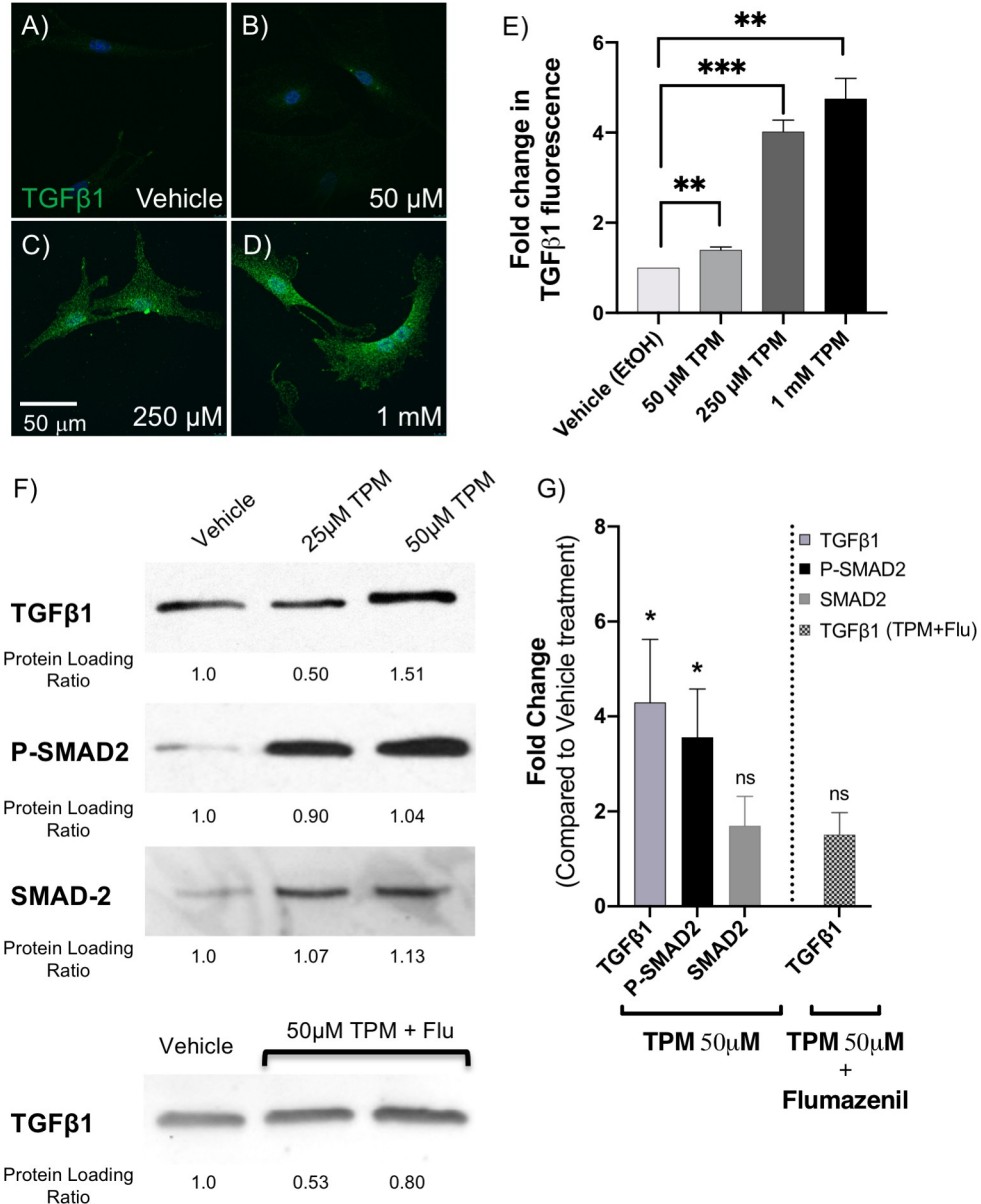

**Fig 2. Topiramate treatment of primary MEPM cells upregulated TGFβ1 expression via GABA receptors.** To validate upregulation of TGFβ1, we isolated primary mouse embryonic palate mesenchyme (MEPM) cells from E13.5 embryos and treated them with 25μM, 50μM, 250μM or 1mM topiramate for 6 hours, as indicated. These cells were analyzed for TGFβ1 expression by immunostaining (A-E) and Western blotting (F-G). There was a significant increase in intracellular TGFβ1 expression upon as little as 50μM topiramate treatment (A-E), which was quantitated in at least 30 cells per treatment from 3 independent experiments (**, p<0.003; ***, p<0.0003). The scale bar represents 50μm (A-D). Western blot analysis also showed a significant increase upon treatment with 50μM topiramate (F, G). We also showed increased phospho-SMAD2 (P-SMAD2) levels with both 25μM and 50 μM topiramate treatments (F, G), indicating an upregulation of TGFβ1 signaling cascade (*, p<0.019). No significant changes were observed in total SMAD-2 level upon 50 μM topiramate treatment (F, G). The upregulation of TGFβ1 protein expression upon 50 μM topiramate treatment is blunted in the presence of 10 μM GABA receptor inhibitor, Flumazenil (F, G).

translocate to the nucleus. Indeed, P-SMAD2 levels were increased following both 25μM and 50μM topiramate treatment (Fig 2F and 2G), indicating a strong upregulation of TGFβ signaling via SMAD-dependent pathway.

## Topiramate treatment increased TGFβ1 protein level via GABA receptors

Since topiramate has been reported to upregulate GABA levels and GABA$_A$ receptor-based signaling [28], which in turn has been shown in other systems to increase TGFβ levels [29, 30], we wanted to see if the same was true in MEPM cells. Thus, we treated the MEPM cells with 50μM topiramate in the presence of 10μM flumazenil, a selective inhibitor of GABA$_A$ receptors [31]. Indeed, flumazenil was able to blunt the effect of topiramate on TGFβ1 level (Fig 2F and 2G), indicating that topiramate was mediating its effect through GABA$_A$ receptors.

## Topiramate treatment of primary MEPM cells results in increased SOX9 expression

After validating the involvement of TGFβ signaling, we wanted to look at a TGFβ target gene with a role in orofacial clefting. We decided to look at expression of SOX9, a transcription factor belonging to the SOXE group with a key role in regulating chondrocyte function [32, 33]. *SOX9* mutations have been identified in patients with Pierre Robin sequence [34] and campomelic (or acampomelic) dysplasia with or without sex reversal [35, 36]. Pierre Robin sequence is a series of defects including small jaw, a posteriorly placed tongue and cleft palate [34]. Campomelic dysplasia is an autosomal dominant skeletal disorder that is characterized by bowed limbs, hypoplastic or hypomineralized bones, and small chest size. Cleft palate, micrognathia (including Pierre Robin sequence), flat face and hypertelorism are also frequent features of Campomelic dysplasia. Interestingly, *SOX9* mutations in patients with Pierre Robin sequence were located in the regulatory region affecting *SOX9* expression. Furthermore, both knockdown and overexpression of *Sox9* in mouse has been shown to result in cleft palate phenotypes [32, 33, 37]. Therefore, an effect on SOX9 expression downstream of TGFβ1 in our system would represent a plausible pathogenetic mechanism of topiramate-based facial clefts. Indeed, we found that SOX9 expression is increased following topiramate treatment (Fig 3A and 3B), which is consistent with a role for TGFβ1 in stabilizing SOX9 protein [38]. Expression of another SOXE transcription factor group member associated with orofacial clefts, SOX10, is not altered upon topiramate treatment (Fig 3A and 3C), showing specificity for SOX9

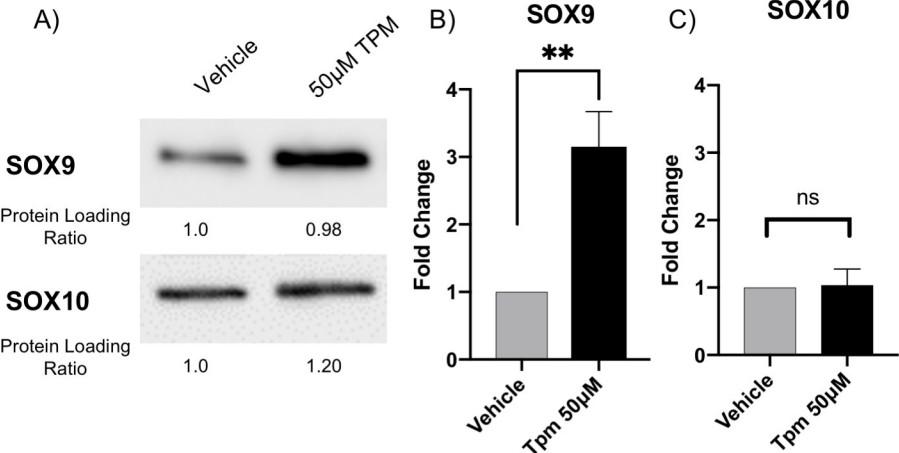

**Fig 3. Topiramate treatment of primary MEPM cells resulted in increased SOX9 expression.** We looked at expression of SOX9, a TGFβ1 target gene involved in orofacial clefting. Western blot analysis of topiramate-treated primary MEPM cells resulted in statistically significant increase in SOX9 expression (A,B; **, p<0.006). In contrast, expression of SOX10 is not altered upon topiramate treatment (A,C; ns, not significant).

upregulation. Our results with topiramate suggest that TGFβ1-mediated altered expression of SOX9 underlies the teratogenicity of this antiepileptic drug.

## Discussion

We utilized an unbiased cell signaling antibody array to identify 40 proteins with altered expression following topiramate treatment of HEPM cells. Although a smaller number of proteins were analyzed with the antibody array compared to an RNA-based analysis, our data suggest that the protein-based analysis provided a clearer view of signaling changes. These 40 proteins included TGFβ1, which was significantly upregulated and was the only one associated with cleft palate.

During palatogenesis, all three TGFβ ligands—TGFβ1, TGFβ2 and TGFβ3—are expressed in the palatal shelves. TGFβ1 is strongly expressed in the distal pre-fusion palatal shelves [39]. Therefore, we utilized primary MEPM cells to validate TGFβ1 increase upon topiramate treatment. In humans, increase in TGFβ signaling has been identified in Marfan and Loeys-Dietz syndromes [40–42]. Marfan syndrome patients show overgrowth of the long bones of the arms and legs, and have defects in multiple organ systems including cardiovascular, craniofacial and ocular anomalies. Individuals with Loeys-Dietz syndrome also have cardiovascular and skeletal abnormalities that are more severe than in Marfan syndrome. Craniofacial defects in Loeys-Dietz syndrome patients include craniosynostosis and cleft palate. TGFβ1 has been shown to be a potent inducer of growth inhibition in various cell types [43–45]. However, in palate development, some evidence suggests TGFβ ligands can promote cell proliferation [46]. Regardless of the complex function of individual TGFβ pathway molecules, perturbation of TGFβ signaling affects many genes associated with orofacial clefts [46, 47] that can negatively impact palatogenesis.

To show that TGFβ1 increase can affect downstream genes associated with orofacial clefts, we looked at expression of SOX9 transcription factor. Deficiency of SOX9 leads to Campomelic dysplasia, characterized by facial and skeletal anomalies, including cleft palate, midface hypoplasia, short stature and short and bowed limbs [35, 36]. However, we actually observed an increase in SOX9 levels following topiramate treatment. TGFβ-mediated increase in SOX9 expression has also been recently shown in mesenchymal fibroblasts to promote renal fibrosis [48]. Importantly, when SOX9 is overexpressed in all chondrocytes, by insertion of *Sox9* cDNA into the *Col2a1* locus, it also results in cleft palate phenotype [32]. Thus, perturbation of SOX9 expression in either direction affects palatogenesis. A sudden surge in TGFβ1 level would therefore disrupt this delicate regulation of SOX9 expression in the palate mesenchyme, and thus result in OFC.

Phosphorylation of SMAD2, a downstream effector of the TGFβ pathway, is necessary for the stabilization of *Sox9* in palatogenesis [38]. Phospho-SMAD2 was upregulated in topiramate-treated MEPM cells, validating that topiramate increases TGFβ signaling cascade. In a recent study examining the effects of topical topiramate on wound healing using mice found significantly increased levels of TGFβ and SOX2 in epidermal cells treated with topiramate [49], which is consistent with a topiramate-TGFβ-SOX cascade observed in our analysis.

Antiepileptic drugs ameliorate CNS excitatory seizures by dampening overall neuronal activity. This includes downregulation of excitatory signals from glutamate receptors and upregulation of inhibitory signals from GABA receptors [50]. Topiramate has been reported to upregulate GABA levels and $GABA_A$ receptor-based signaling [28]. Importantly, increased GABA signaling upregulates TGFβ levels [29, 30]. While mouse mutants for some components of the GABA signaling pathway result in cleft palate phenotype [51–53], the effect of GABA upregulation on palatogenesis has not been explored. Recently, upregulation of GABA

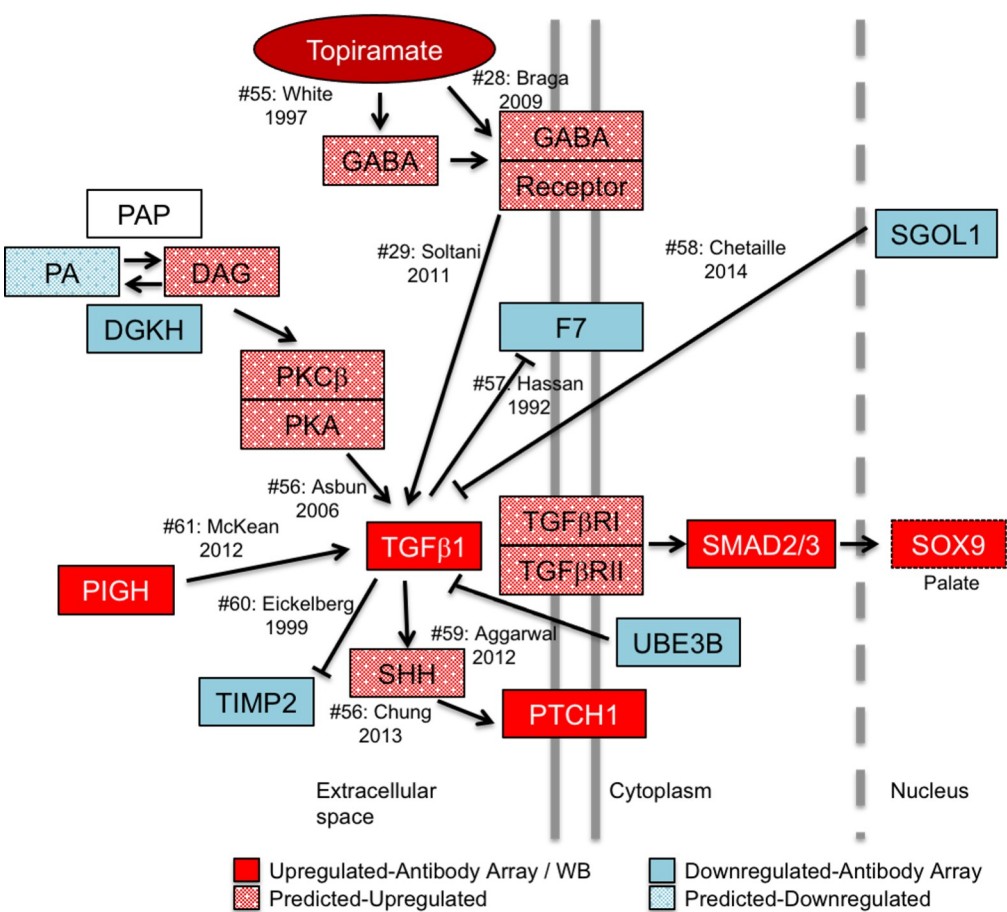

**Fig 4. Model of topiramate action on MEPM cells.** Our model predicts that topiramate is capable of stimulating GABA receptors in the palate to upregulate TGFβ1 expression. TGFβ1 expression is tightly regulated during craniofacial morphogenesis. Dysregulation of TGFβ1 signaling can lead to altered expression of genes associated with orofacial clefts. For example, we show that topiramate leads to upregulation of SOX9 expression, which is sufficient to result in cleft palate in mice. In order to gather corroborating evidence for the upregulation of TGFβ signaling from the 39 (excluding TGFβ) differentially expressed gene products, we performed a manual curation of the literature for connectivity to TGFβ1. Our analysis revealed that changes in an additional seven (18%) of the proteins are consistent with upregulation of TGFβ1 signaling. Moreover, these molecules affect TGFβ1 signaling both upstream and downstream of the TGFβ1 ligand, suggesting a concerted global upregulation of the pathway. The solid-colored molecules are from the antibody-array or western blot results, while the spotted molecules are changes in upstream effectors predicted from literature. Citation number (#) corresponds to listed references.

receptor activity was shown to decrease cell proliferation of embryonic and neural crest cells, as well as of blastocysts [54]. These results are consistent with a potential inhibitory effect of GABA upregulation on overall embryonic development and, in particular, on neural crest-influenced orofacial morphogenesis [18]. Also in support, several studies indicate tissue-specific activation of cell death following upregulation of TGFβ signaling, as reviewed in Schuster and Krieglstein [45]. Thus, our data suggest that topiramate-based upregulation of GABA signaling can increase TGFβ signaling, which in turn results in tissue-specific changes in expression of downstream genes, such as SOX9, involved in palatogenesis.

To assess the impact of the genes encoding the 40 differentially expressed proteins on orofacial clefting, we assayed their connectivity to the 107 known OFC-related genes reported by IPA. A single IPA connectivity network included over 55% of both sets of genes (Fig 1). This high level of connectivity argues that exposure to topiramate has the potential to perturb many

OFC-related genes and pathways in the developing palate. The highest connectivity in this network was centered on TGFβ1 ligand. We also considered the possibility that the upregulation extended beyond TGFβ1 ligand to the entire pathway. We reasoned that if TGFβ signaling was indeed upregulated in topiramate-treated HEPM cells, we would find evidence for upregulation of downstream factors from our antibody array data. Indeed, a manually curated network showed evidence for upregulation of TGFβ signaling both upstream and downstream of TGFβ1 ligand (Fig 4) [28, 29, 55–62]. We were able to directly confirm nine molecules that were consistent with upregulation of TGFβ signaling (Fig 4: PIGH, PTCH1, SMAD2/3, SOX9, DGKH, F7, SGOL1, TIMP2, UBE3B). We also found literature support for the topiramate-based increase in GABA signaling [28, 55] to in turn upregulate levels of TGFβ1 [29], which is consistent with our result showing flumazenil blunts this effect. Together, this network not only shows a broad upregulation of the TGFβ signaling pathway following topiramate treatment of HEPM cells, but also suggests perturbation of other signaling pathways important in embryogenesis, including Phospholipase-C-based PKC/PKA and Sonic hedgehog signaling (Fig 4).

The genetic network affected by topiramate treatment of palate mesenchyme cells provides an important framework to study the OFC-related teratogenic effects. Future *in vitro* and *in vivo* studies are required to elucidate the precise role of increased TGFβ signaling in the teratogenicity of topiramate. Topiramate is one of many anti-epileptic and mood-stabilizing drugs that have recently been associated with structural birth defects. All of these drugs share certain molecular characteristics and targets. Therefore, understanding the molecular genetic mechanism behind topiramate teratogenicity will likely also provide clues to the general link reported among a broad class of antiepileptic drugs and birth defects.

## Supporting information

**S1 Fig. Topiramate treatment of HEPM and MEPM cells did not increase TGFB1 transcript level.** A) RNA from HEPM cells treated for 6 hours with vehicle (control) and 1mM Topiramate (Tpm) was analyzed for *TGFB1* expression using qRT-PCR. No increase was observed upon treatment. Data show average and SEM from 4 pairs of treatments. B) RNA from MEPM cells treated for 6 hours with vehicle and 50uM Tpm was analyzed for *Tgfb1* expression using qRT-PCR. Data are shown from 5 experiments. No significant change was observed between vehicle and treatment. ns = not significant.
(TIF)

**S1 Table. List of all 57 proteins with altered expression in topiramate-treated HEPM cells.**
(TIF)

**S2 Table. List of IPA predicted OFC-related genes.**
(TIF)

**S3 Table. IPA predicted diseases and bio-functions associated with the 40 gene-products significantly altered in topiramate-treated HEPM cells.**
(TIF)

**S4 Table. IPA predicted networks associated with the 40 gene-products significantly altered in topiramate-treated HEPM cells.**
(TIF)

**S1 Raw images.**
(PDF)

## Acknowledgments

We thank Dr. Partha Kasturi and Dr. Hemantkumar Chavan (KUMC) for assistance and use of their equipment; Dr. Karin Zueckert-Gaudenz and Mr. Brian Flaherty (Molecular Biology Core, Stowers Institute for Medical Research, Kansas City, MO) for use of their microarray scanners; Ms. Shannon Zhang (Full Moon BioSystems Inc.) for assistance with antibody array data analysis; Mr. Byunggil Yoo (KUMC) for assistance and useful discussions.

## Author Contributions

**Conceptualization:** Syed K. Rafi, Irfan Saadi.

**Data curation:** Syed K. Rafi, Irfan Saadi.

**Formal analysis:** Syed K. Rafi, Jeremy P. Goering, Adam J. Olm-Shipman, Lauren A. Hipp, Nicholas J. Ernst, Nathan R. Wilson, Everett G. Hall, Sumedha Gunewardena, Irfan Saadi.

**Funding acquisition:** Irfan Saadi.

**Investigation:** Syed K. Rafi, Jeremy P. Goering, Adam J. Olm-Shipman, Lauren A. Hipp, Nathan R. Wilson, Everett G. Hall, Sumedha Gunewardena, Irfan Saadi.

**Methodology:** Syed K. Rafi, Jeremy P. Goering, Adam J. Olm-Shipman, Lauren A. Hipp, Nicholas J. Ernst, Nathan R. Wilson, Everett G. Hall, Sumedha Gunewardena, Irfan Saadi.

**Project administration:** Irfan Saadi.

**Resources:** Sumedha Gunewardena, Irfan Saadi.

**Software:** Adam J. Olm-Shipman, Sumedha Gunewardena.

**Supervision:** Irfan Saadi.

**Validation:** Syed K. Rafi, Jeremy P. Goering, Adam J. Olm-Shipman, Lauren A. Hipp, Nicholas J. Ernst, Nathan R. Wilson, Sumedha Gunewardena, Irfan Saadi.

**Visualization:** Syed K. Rafi, Jeremy P. Goering, Adam J. Olm-Shipman, Lauren A. Hipp, Nicholas J. Ernst, Nathan R. Wilson, Everett G. Hall, Sumedha Gunewardena, Irfan Saadi.

**Writing – original draft:** Syed K. Rafi, Jeremy P. Goering, Irfan Saadi.

**Writing – review & editing:** Syed K. Rafi, Jeremy P. Goering, Adam J. Olm-Shipman, Lauren A. Hipp, Nicholas J. Ernst, Nathan R. Wilson, Everett G. Hall, Sumedha Gunewardena, Irfan Saadi.

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
