## [Decision Letter · Decision Letter 0]

21 Oct 2020

PONE-D-20-27474

Anti-epileptic drug topiramate upregulates TGFβ1 and SOX9 expression in primary embryonic palatal mesenchyme cells: Implications for teratogenicity

PLOS ONE

Dear Dr. Saadi,

Thank you for submitting your manuscript to PLOS ONE. After careful consideration, we feel that it has merit but does not fully meet PLOS ONE’s publication criteria as it currently stands. Therefore, we invite you to submit a revised version of the manuscript that addresses the points raised during the review process.

The reviewers indicate that the manuscript has scientific merit, but they suggest some important revisions to the text.  Also, reviewers suggested experiments to measure TGF-beta mRNA levels using quantitative PCR and to measure SMAD2 protein levels for comparison to p-SMAD2 levels.  These experiments will improve the study, making the conclusions more complete.

We look forward to receiving your revised manuscript.

Kind regards,

James A. Marrs

Academic Editor

PLOS ONE

Additional Editor Comments:

The reviewers suggest some important revisions to the text, which should be addressed. Also, several experiments are suggested. Validating the effects of the pathways would involve several experiments. Although these are good suggestions, the scientific contribution without these experiments is reasonable. The other reviewer suggests some quantitative PCR to measure TGF-beta mRNA levels and measurement of SMAD2 protein levels for comparison to p-SMAD2 levels. I recommend these experiments to improve the scientific completeness of the study.

Journal Requirements:

2. Please provide the product number and any lot numbers of the antibodies used for Western blotting.

3. At this time, we ask that you please provide scale bars on the microscopy images presented in Figure 2 and refer to the scale bar in the corresponding Figure legend.

4. Please provide additional information about each of the cell lines used in this work, including any quality control testing procedures (authentication, characterisation, and mycoplasma testing). For more information, please see http://journals.plos.org/plosone/s/submission-guidelines#loc-cell-lines.

5. At this time, we request that you  please report additional details in your Methods section regarding animal care, as per our editorial guidelines: 1) Please provide details of the number of animals used in this study, and the source of the mice, 2) please provide details of animal welfare (e.g., shelter, food, water, environmental enrichment), and 3) please describe the age of the embryos and the method of sacrifice. Thank you for your attention to these requests.

Reviewers' comments:

Reviewer's Responses to Questions

**Comments to the Author**

1. Is the manuscript technically sound, and do the data support the conclusions?

Reviewer #1: Yes

Reviewer #2: Yes

2. Has the statistical analysis been performed appropriately and rigorously? 

Reviewer #1: Yes

Reviewer #2: I Don't Know

3. Have the authors made all data underlying the findings in their manuscript fully available?

Reviewer #1: Yes

Reviewer #2: Yes

4. Is the manuscript presented in an intelligible fashion and written in standard English?

Reviewer #1: Yes

Reviewer #2: Yes

5. Review Comments to the Author

Reviewer #1: This article describes an antibody screen to find downstream targets of topiramate in palatial mesenchymal cells that may be causative in palatial clefting. The manuscript is well written and easy to follow. The procedures and statistics all seem to have been performed appropriately, and the results are interesting, as far as they go. It’s just that the study doesn’t go very far. It starts out strong by identifying target genes and pathways, but then it stops before doing anything with the results beyond bioinformatics and speculative model building.

For example, the authors state, “ To validate that TGFb1 increase plays a physiological role in the etiology of OFC, we utilized

primary MEPM cells.” But, that’s not really what they did. All they did was look at gene and protein increases in related cells. The same thing with SOX9, they just showed up regulation and Smad phosphorylation. This is a good start, but why not do an experiment to show a change in the physiology of the cells? They have both a plate mesenchyme cell line and primary palate cells. Both are easy to culture, so why not look for things like changes in proliferation or apoptosis as a consequence of topiramate? The pathway analysis they show predicts that topiramate action through those genes changes cell survival and growth (e.g: see Table S3). Then they could do straightforward knockdown or inhibitor experiments to change Tgfb1 activity or SOX9 protein levels during topiramate treatment. These simple experiments would go a long way toward validating a potential physiological role for these pathways in topiramate action in vivo. It’s fairly easy to draw line diagrams of pathways and build models on paper, but which potential mechanisms are really important to the biology being studied? In other words, how important is Tgfb1 really to the teratogenic effect of topiramate compared to all the other possibilities?

The same principle is true with the speculation about GABA involvement in activating Tgfb1 expression. It’s easy to speculate, but it should also be easy to test. Why not do cell culture experiments with and without GABA inhibitor to see if it changes Tgfb1 levels?

On another subject, I am not clear about why SOX10 expression is a useful control for the specificity of the SOX9 upregulation. Is there some scenario of generalized transcriptional upregulation that involves SOX10 such that it needs to be controlled for and this gene is a good indicator or that? Are SOX9 and 10 genes regulated together in other systems such that this connection needs to be ruled out here?

I have one final general statement about the organization of the manuscript relating to concision. It seems to me that the Discussion section spends a lot of words restating the results from the previous section, such that they almost seem to blend together. I would recommend that the authors spend some time rewriting the Discussion to be more separate from the Results, more concise, and a little less rambling. For example, the second paragraph of the Discussion goes on for several lines about the three Tgfbeta ligands, their expression in palate, and their possible effects on proliferation of palatal cells. This could really be replaced by a couple of sentences about Tgfb1 and what specifically is known about its effect in palate. That is really what the reader needs to evaluate the importance of the results presented in this study.

Reviewer #2: Topiromate TGF-beta review

Exposure to Topiromate in utero is associated with increased incidence of cleft palate among other craniofacial defects however the mechanism by which topiromate affects craniofacial development has not been understood. Here, Rafi et al. use an antibody array to identify changes in protein concentrations in primary palate cultures that are exposed to Topiromate. TGF-beta1 is identified as an overexpressed protein with exposure to topiromate. The authors show topiromate increases phosphorylation of Smad2, a readout of TGF-beta signaling. They also show that TGF-beta target gene (Sox9) expression is increased upon topiromate exposure. This work is important because topiromate is a commonly used medication to treat epilepsy, migraine, and bi-polar disorder. It is important that clinicians and patients understand the risks associated with the medication and the mechanism by which the medication may disrupt development.

Suggestions:

Topiromate is also used as a migraine preventative medication. Because migraines are more common than epilepsy, it is worth referencing this in both the abstract and introduction. Migraine affects almost a third of the female population who are of child bearing age, so this is also really important for the significance of this study.

Add a transition between the sentence “Despite broad use and

75 teratogenic potential of topiramate, the molecular mechanism(s) underlying the increased76 occurrence of major congenital malformations is not understood (6, 7, 10, 12, 16).” And the next paragraph.

(SOX9), “a known 92 clefting gene and TGFB1 target.”- this phrase is a little unclear. It would be more clear to say “a TGFB1 target that causes causes cleft palate when overexpressed”- or something that states overexpression causes cleft palate rather than loss of function.

Please refrain from using acronyms when possible, for example “OFC” could be replaced with “gene associated with oral or facial clefts. This would make the text more accessible to the general readership of this journal.

Cell cycle and apoptosis genes were also identified in the IPA. Increased apoptosis and decreased cell proliferation could also cause clefting. The focus on TGF-beta is great, but the authors also could have focused on a direct mechanism by looking at cell proliferation and apoptosis.

This is interesting: “We saw a significant increase in intracellular TGF�1 level at

246 the physiological 50μM topiramate treatment (Figure 2E).” We don’t expect TGFb1 to function intracellularly. Was there not an increase in TGFb1 in the media, extracellular portion? Or did the authors not check? IF the authors have not checked, I suggest checking and comparing TGF-beta in the media as well.

Figure 2A-E: The representative cells for 250 uM topiromate show a clear increase in fluorescence. However, D, 1mM topiromate looks like it has less fluorescence than C or B. If there is indeed a greater fluorescence in most cells treated with 1mM topiromate, please find a different representative image. The apparent discrepancy between fluorescence in figure 2D and the quantification in E made me wonder about how the quantification of fluorescence was measured. There are no details about whether the fluorescence is quantified and normalized to the number of cells or if each individual cell is outlined as a region of interest. If the fluorescence is not normalized to the number of cells in an image in any way, the fluorescence quantification is questionable. These details should be provided to allow the reader to interpret the analysis.

I would be interested to see if the upregulation of TGF-beta1 is due to an increase in transcription of TGF-beta1 on the mRNA level or only at the level of protein. Please perform Q-RT PCR to determine if mRNA levels of the ligand are upregulated

Were P-Smad2/3 levels from the nuclear fraction or total cell lysates? Also, were Smad2 protein levels also quantified? It would be interesting to know if Smads are also upregulated or if only the phosphorylation of Smad2/3 are increased with the topiromate treatment.

Consider changing color scheme from Red and Green in Figure 4 to colors that are color-blind friendly.

6. PLOS authors have the option to publish the peer review history of their article (what does this mean?). If published, this will include your full peer review and any attached files.

Reviewer #1: No

Reviewer #2: No

---

## [Author Response · Author response to Decision Letter 0]

4 Jan 2021

please see attached file: Response to Reviewers

---

## [Decision Letter · Decision Letter 1]

1 Feb 2021

Anti-epileptic drug topiramate upregulates TGFβ1 and SOX9 expression in primary embryonic palatal mesenchyme cells: Implications for teratogenicity

PONE-D-20-27474R1

Dear Dr. Saadi,

We’re pleased to inform you that your manuscript has been judged scientifically suitable for publication and will be formally accepted for publication once it meets all outstanding technical requirements.

Kind regards,

James A. Marrs

Academic Editor

PLOS ONE

Additional Editor Comments (optional):

The two reviewers have evaluated the manuscript revision. The 2nd reviewer initially had trouble finding the response to reviewers and the red-lined version of the manuscript. I shared these documents with the reviewer, and the reviewer changed their recommendation to "accept".

Thank you for your efforts to respond to the reviewers and for the additions to the manuscript. Congratulations on your contribution.

Reviewers' comments:

Reviewer's Responses to Questions

**Comments to the Author**

1. If the authors have adequately addressed your comments raised in a previous round of review and you feel that this manuscript is now acceptable for publication, you may indicate that here to bypass the “Comments to the Author” section, enter your conflict of interest statement in the “Confidential to Editor” section, and submit your "Accept" recommendation.

Reviewer #1: All comments have been addressed

Reviewer #2: (No Response)

2. Is the manuscript technically sound, and do the data support the conclusions?

Reviewer #1: Yes

Reviewer #2: Yes

3. Has the statistical analysis been performed appropriately and rigorously? 

Reviewer #1: Yes

Reviewer #2: I Don't Know

4. Have the authors made all data underlying the findings in their manuscript fully available?

Reviewer #1: Yes

Reviewer #2: No

5. Is the manuscript presented in an intelligible fashion and written in standard English?

Reviewer #1: Yes

Reviewer #2: Yes

6. Review Comments to the Author

Reviewer #1: (No Response)

Reviewer #2: I could not find any response to reviewers' comments nor could I find a manuscript with changes highlighted as requested by the journal. I did not see any addressing of reviewer 1's good suggestions to follow up looking at apoptosis or proliferation. While there was an additional experiment to look at whether TGF beta1 was upregulated at the transcriptional or translational level, this figure was relegated to supplemental and not discussed enough in the text. It is really interesting that there was no change in transcriptional levels of TGFbeta, but there was an increase in protein. This might also have something to do with TGFbeta that is released into the media vs. stuck in cells when topiromate is applied. Again, this would have been addressed with an ELISA looking at TGFbeta in the media as I suggested in my first review. The study is a great start and there is something interesting here to follow up. But the manuscript does not go far enough into mechanism (either by doing the experiments I suggested or doing those that reviewer 1 suggested or both) to merit publication in Plos One. I would suggest publishing it in one of the the other plos journals that doesn't have a reputation for including mechanism and more depth.

7. PLOS authors have the option to publish the peer review history of their article (what does this mean?). If published, this will include your full peer review and any attached files.

Reviewer #1: No

Reviewer #2: No

---

## [Editor Report · Acceptance letter]

3 Feb 2021

PONE-D-20-27474R1 

Anti-epileptic drug topiramate upregulates TGFβ1 and SOX9 expression in primary embryonic palatal mesenchyme cells: Implications for teratogenicity 

Dear Dr. Saadi:

I'm pleased to inform you that your manuscript has been deemed suitable for publication in PLOS ONE. Congratulations! Your manuscript is now with our production department. 

Kind regards, 

on behalf of

Dr. James A. Marrs 

Academic Editor

PLOS ONE